# Early Risk Score for Predicting Hypotension in Normotensive Patients with Non-Variceal Upper Gastrointestinal Bleeding

**DOI:** 10.3390/jcm8010037

**Published:** 2019-01-02

**Authors:** Byuk Sung Ko, Youn-Jung Kim, Dae Ho Jung, Chang Hwan Sohn, Dong Woo Seo, Yoon-Seon Lee, Kyoung Soo Lim, Hwoon-yong Jung, Won Young Kim

**Affiliations:** 1Department of Emergency Medicine, College of Medicine, Hanyang University, 222 Wangsimni-ro, Seongdong-gu, Seoul 133-791, Korea; postwinston@gmail.com; 2Department of Emergency Medicine, Asan Medical Center, College of Medicine, University of Ulsan, 88 Olympic-ro 43-gil, Songpa-gu, Seoul 05505, Korea; yjkim.em@gmail.com (Y.-J.K.); faraday520@gmail.com (D.H.J.); schwan97@gmail.com (C.H.S.); leiseo@gmail.com (D.W.S.); ysdoc@amc.seoul.kr (Y.-S.L.); kslim@amc.seoul.kr (K.S.L.); 3Research Scholar, Department of Biomedical Informatics, School of Medicine, University of San Diego, 9500 Gilman Drive #0728, La Jolla, CA 92093, USA; 4Department of Gastroenterology, Asan Medical Center, College of Medicine, University of Ulsan, 88 Olympic-ro 43-gil, Songpa-gu, Seoul 05505, Korea; hwoonymd@gmail.com

**Keywords:** non-variceal upper gastrointestinal bleeding, hypotension, lactate, risk model

## Abstract

Risk assessment for upper gastrointestinal bleeding (UGIB) is important; however, current scoring systems are insufficient. We aimed to develop and validate a prediction model for rapidly determining the occurrence of hypotension in non-variceal UGIB patients with normotension (systolic blood pressure ≥90 mmHg) at emergency department presentation. In this prospective observational cohort study, consecutive non-variceal UGIB patients between January 2012 and April 2017 were enrolled. We developed and validated a new prediction model through logistic regression, with the occurrence of hypotension <24 h as the primary outcome. Among 3363 UGIB patients, 1439 non-variceal UGIB patients were included. The risk factors for the occurrence of hypotension were lactate level, blood in nasogastric tube, and systolic blood pressure. The area under the curve (AUC) of the new scoring model (LBS—Lactate, Blood in nasogastric tube, Systolic blood pressure) in the development cohort was 0.74, higher than the value of 0.64 of the Glasgow–Blatchford score for predicting the occurrence of hypotension. The AUC of the LBS score in the validation cohort was 0.83. An LBS score of ≤2 had a negative predictive value of 99.5% and an LBS score of ≥7 had a specificity of 97.5% in the validation cohort. The new LBS score stratifies normotensive patients with non-variceal UGIB at risk for developing hypotension.

## 1. Introduction

Although the morbidity and mortality of upper gastrointestinal bleeding (UGIB) have decreased recently, this condition remains a burden to public health, with a mortality rate of 6–12% and hospital costs of more than US$2.5 billion yearly in the United States [1]. The American College of Gastroenterology and international consensus guidelines recommend early risk stratification of non-variceal UGIB patients to stratify them into higher and lower risk categories, which may assist in initial decisions such as the timing of endoscopy, time of discharge, and level of care (e.g., ward vs. step-down vs. intensive care) [2]. Several risk scoring systems such as the Glasgow–Blatchford score (GBS) and the Rockall score have been developed for assessing patients with UGIB, and they are useful tools for identifying low-risk patients (especially the GBS). However, they have limitations in identifying high-risk patients who will require inpatient endoscopy, embolization, and surgical treatment, and in identifying patients at high risk for hemodynamic instability [3,4,5,6,7]. Moreover, the subjectivity of the definitions of hepatic disease and cardiac disease included in the GBS makes its application in clinical practice difficult.

Hemodynamic instability, such as the development of hypotension, is known as a predictive factor for rebleeding or mortality in non-variceal UGIB [2,8,9], and is a more objective outcome variable than those reported in previous studies, which include the requirements for endoscopic hemostasis, clinical intervention, and transfusion. Early determination of the severity of UGIB is important for optimizing care and efficiently allocating resources; however, for patients presenting with stable vital signs, it is challenging to decide whether to urgently perform endoscopy and admit potentially high-risk patients to intensive care or progressive care units, and there is a paucity of information on the risk stratification of hemodynamically stable UGIB patients at admission. To address this issue, our group previously published a study that confirmed that the initial lactate level, which is a useful surrogate measure of inadequate tissue perfusion, can predict the in-hospital occurrence of hypotension in stable patients with acute non-variceal UGIB [10]. The aim of our study was to develop and validate a prediction model including lactate for rapidly determining the occurrence of hypotension in non-variceal UGIB patients with normotension at initial presentation.

## 2. Materials and Methods

### 2.1. Study Population

A prospective observational cohort study was conducted in the emergency department (ED) of the Asan Medical Center, which has a census record of 110,000 visits per year and serves as a tertiary referral center in Seoul, Korea. All non-variceal UGIB patients older than 18 years visiting the ED were enrolled in the UGIB registry. We defined UGIB by chief complaint of hematemesis, coffee-ground vomiting, or melena [11]. Patients whose lactate level was not measured, liver cirrhosis patients who may have had variceal bleeding, and advanced neoplasm patients whose lactate level could be affected were also excluded. Patients with hypotension, defined as a systolic blood pressure (SBP) of <90 mmHg at ED triage, were excluded because we aimed to investigate a risk model for normotensive patients with non-variceal UGIB.

### 2.2. Evaluation at Presentation

We used the variables of the UGIB registry to derive our prognostic model. The variables in the registry were collected on the basis of previous risk prediction studies, which are routinely available at ED presentation. These included demographics, initial vital signs, comorbidities, mental change, syncope, fresh blood in nasogastric tube, melena on rectal examination, specific medications that could cause gastrointestinal bleeding (non-steroidal anti-inflammatory drugs, antiplatelet agents, anticoagulants), hemoglobin level, platelet count, prothrombin time, international normalized ratio, blood urea nitrogen level, creatinine level, albumin level, base deficit, lactate level, GBS, pre-endoscopy Rockall score, and hypotension development. Lactate levels were measured in arterial or venous blood using a point-of-care testing blood gas analyzer (GEM Premier 3500 with iQM, A Werfen Company, Bedford, MA, USA), which detects a lactate level range of 0.3–15.0 mmol/L within 10 min of ED presentation. Lactate level can be assessed within 1 min by using the blood gas analyzer.

### 2.3. Management and Follow-Up

Nasogastric tube and rectal examinations were performed in all patients except in those who refused or did not cooperate well, to identify significant bleeding in the upper gastrointestinal tract. Pre-endoscopic intravenous proton pump inhibitors (PPIs) were used routinely in all patients until there was no evidence of peptic ulcer disease. Continuous infusion PPI therapy was used to decrease rebleeding and mortality in patients with high-risk stigmata who had undergone endoscopic hemostasis. Somatostatin and octreotide were not routinely used unless there was evidence of variceal bleeding. Blood transfusion was considered for use in patients with shock, hypotension, clinical deterioration, or Hgb less than 7 g/dL. Transfusion for coagulopathy or thrombocytopenia was performed at the discretion of the treating physician. Endoscopy was conducted within 24 h in all cases, except in patients who refused to undergo endoscopy or who were in a too poor condition to undergo the procedure. Other treatments such as fluids, transfusions, or proton pump inhibitors were provided to those patients identically with patients who received endoscopy. The specific endoscopic hemostasis method was decided at the discretion of the endoscopist. Routine second-look endoscopy was not conducted. Follow-up endoscopy was conducted in patients with a high risk for rebleeding on first endoscopy or in those suspected to have rebleeding based on vital signs, hemoglobin change, or other clinical judgments. Patients who were at a low risk for rebleeding on the basis of endoscopic criteria and clinical judgment were discharged and managed as outpatients. Patients with high-risk stigmata were hospitalized for at least 3 days. All patients with peptic ulcers were tested for *Helicobacter pylori* and received eradication therapy in the case of a positive result. The ED at the Asan Medical Center has a protocol in which blood pressure is measured after 15 min if hypotension occurs in patients with UGIB.

### 2.4. Definition and Outcomes

We judged patients as having mental change depending on their response to mental status assessment (verbal, response to pain, or unresponsive) at ED triage according to the AVPU (alert, voice, pain, unresponsive) scale. We judged the presence of fresh blood in nasogastric tube when fresh blood was seen on nasogastric tube irrigation. If fresh blood color on nasogastric tube was still seen after manual irrigation with 500 mL of distilled water, we judged that case as positive. Melena was defined as present when black or tarry feces were observed on rectal examination. Concerning comorbidities, chronic liver disease was defined as a diagnosis of chronic hepatitis B, hepatitis C, alcoholism, autoimmune disease, or others. Coagulopathy was defined as present if the patient’s baseline platelet count and prothrombin time were outside of the normal range (normal ranges of prothrombin time (%) and platelets are 70–140% and 150,000–350,000/mm^3^, respectively). Ischemic heart disease was defined as angiographically proven coronary artery disease. Heart failure was defined as an evidence of decreased ejection fraction on echocardiography or clinically diagnosed heart failure. Syncope was defined as present when the patient experienced loss of consciousness. Advanced neoplasm was defined as a confirmed distant metastasis along with primary cancer. The primary outcome was the development of hypotension, which was defined as an SBP of <90 mmHg for >15 min without signs of other causes of hypotension except blood loss, within 24 h of ED presentation [10].

### 2.5. Development and Validation of the New Prognostic Model

This study included consecutive patients with non-variceal UGIB who presented to the ED between 1 January 2012, and 30 April 2017. Among them, we selected patients in time order from 1 January 2012 to 31 December 2015 for the development set and from 1 January 2016 to 30 April 2017 for the validation set. We developed our prediction model by using logistic regression, with the occurrence of hypotension as the outcome.

### 2.6. Statistical Analysis

In the model development cohort, univariable and multivariable analyses were performed to investigate the association between various risk factors and the occurrence of hypotension within 24 h. To develop a statistically robust scoring model for predicting the occurrence of hypotension within 24 h, we used a bootstrap resampling method. The logistic regression models were assessed using the Hosmer–Lemeshow goodness-of-fit test and the C-statistic. Sullivan et al. have provided details on the development of scoring systems [12]. Validation of the model was performed separately by measuring discrimination and calibration abilities in the separate validation cohort. First, we calculated the *C*-statistic. Second, the calibration was tested. Calibration was assessed by plotting the predicted probabilities against the actual outcome.

Student’s *t*-test for continuous variables and the χ^2^ test or Fisher’s exact test for categorical variables were used. The associations between the occurrence of hypotension within 24 h and the characteristic variables of participants were tested using univariable and multivariable analyses. The backwards elimination method and clinical significance were used to develop a multivariable model. We compared the area under the curve (AUC) of the prediction model with those of the GBS and the pre-endoscopy Rockall score to assess the discriminatory power of our new model in predicting the occurrence of hypotension. All reported *p*-values were two-sided, and *p* < 0.05 was considered significant. All statistical analyses were performed using SAS version 9.4 (SAS Institute Inc., Cary, NC, USA) and SPSS for Windows version 18.0 (SPSS Inc., Chicago, IL, USA). The Asan Medical Center Institutional Review Board approved this study and waived informed consent due its nature as an observational study.

## 3. Results

During the study period 3363 patients of UGIB was screened. We excluded 1038 patients with liver cirrhosis, 519 patients who presented with low initial SBP, 313 patients with advanced neoplasm, and 54 patients without lactate measurement. Finally, 1439 non-variceal UGIB patients who presented with normotension were included. We selected 1046 consecutive patients (72.7%) for the development set. Validation for this new prognostic model was conducted in 393 (27.3%) patients. There was no significant relevant difference between development and validation cohort except heart rate (mean, 91.5/min vs. 88.1/min), respiratory rate (mean, 19.7/min vs. 19.2/min), prothrombin time/international normalized ratio (PT/INR, 89% vs. 82%), and proportion of chronic liver disease (3.6% vs. 6.8%).

### 3.1. Development of the New Prognostic Model

The univariable analysis model for predicting the occurrence of hypotension in the development set is summarized in Table 1. After the bootstrap resampling method, initial SBP, lactate level, and fresh blood in nasogastric tube were significant factors in the multivariable logistic regression analysis. Table 2 shows the new prognostic models for predicting the occurrence of hypotension in non-variceal UGIB patients. The *C*-statistic for the new prognostic model was 0.735 (95% confidence interval (CI), 0.686–0.784). The Hosmer–Lemeshow test for the calibration of the new prognostic model showed that the model appropriately fit the data with χ^2^ statistics of 5.245 (*p* = 0.513). Whereas a score of 1 showed a 12.1% occurrence of hypotension, scores of 8 and 9 showed a 38.5% and 60.0% occurrence of hypotension, respectively (Figure 1). The performance of the new prognostic model in different thresholds is presented in Table 3. The cutoff value of ≥3 showed 82.5% sensitivity, 53.1% specificity, 15.2% positive predictive value (PPV), and 96.7% negative predictive value (NPV).

Hypotension occurred in 97 patients (9.3%) in development group. Other managements and outcomes of the patients in the development group are summarized in Appendix A. Gastric ulcer was most common on endoscopic finding in the patients in the development group (Appendix A).

### 3.2. Score Comparison in the Development and Validation Sets

The diagnostic performance of the new score model for predicting the occurrence of hypotension, assessed using the AUC, was higher (0.735; 95% CI, 0.686–0.784) than that of the GBS (0.644; 95% CI, 0.590–0.698) and the pre-endoscopy Rockall score (0.524; 95% CI, 0.465–0.583) in the development set (Figure 2). The diagnostic performance for predicting the occurrence of hypotension in the validation set was similar between the new score model and the GBS and the pre-endoscopy Rockall score. The AUC of the new score was 0.833 (95% CI, 0.776–0.890), which is higher than the AUC of the GBS (0.731; 95% CI, 0.652–0.810) and the AUC of the pre-endoscopy Rockall score (0.572; 95% CI, 0.476–0.667) (Figure 3). To test the accuracy in groups classified as high risk, the sensitivity, specificity, PPV, and NPV were examined using a cutoff value of ≤2 in the new score model. Of the 194 episodes classified as high risk by the new score model, 36 episodes (18.6%) showed the occurrence of hypotension, indicating sensitivity, specificity, PPV, and NPV of 97.3%, 55.6%, 18.6%, and 99.5%, respectively (Table 4). When a cutoff value of ≥7 was applied, the specificity of the new score model was 97.5%. The AUC of the new score for predicting the need for red blood cell (RBC) transfusion was 0.698 (95% CI, 0.647–0.750) while that of the GBS was 0.838 (95% CI, 0.798–0.878). The diagnostic performance of the pre-endoscopy Rockall score was 0.607 (95% CI, 0.552–0.663). The median amount of RBC transfusion in the low risk group (≤2 in new score) was 0.0 (0.0–2.0) while for those in the high risk group (≥3) it was 2.0 (0.0–4.0) (*p* < 0.001). The median amount of RBC transfusion in the high risk (≥7) and low risk groups classified by GBS was 0.0 (0.0–0.0) and 2.0 (0.0–3.8), respectively. A significant difference was also observed between the high risk and low risk groups classified by GBS (*p* < 0.001). We examined the accuracy using a cutoff value of 2 in the new score model. The sensitivity, specificity, PPV, and NPV values of the new score for predicting the need for red blood cells were 62.7%, 66.7%, 67.0%, and 62.4%, respectively (Appendix A).

## 4. Discussion

A simple and objective risk score model would be helpful in identifying high-risk UGIB patients with normotension at ED presentation. Through this study, we identified three independent variables for the occurrence of hypotension: lactate level, blood in nasogastric tube, and initial SBP. With these factors, non-variceal UGIB episodes were divided into two different classes according to increasing occurrence of hypotension. Testing the accuracy of the new score showed that it has higher sensitivity and NPV than the GBS. Therefore, it might be a useful tool for triaging high- or low-risk patients with non-variceal UGIB at the point of care.

The optimal threshold for identifying low-risk patients has been recommended in published guidelines as a GBS of 0 [2,13]. However, some authors suggested a GBS of ≤1 as the threshold for identifying low-risk patients [11,14]. These thresholds were established to identify patients at low risk for clinical intervention and select patients who can be safely managed as outpatients without early endoscopy. In contrast, the primary outcome in our study was the development of hypotension; thus, it is difficult to compare our results with those of previous research. Stanley et al. demonstrated that a GBS of ≥7 was the best at predicting the need for endoscopic treatment. The sensitivity and specificity were 80.4% and 57.4%, respectively [11]. In regard to NPV, new score ≤2 was higher than GBS ≥7 (99.5% vs. 92.4%) which could be useful in excluding high risk patients. A new score ≥7 also showed higher PPV than the GBS ≥7 (50.0% vs. 12.3%). Furthermore, Stanley et al. concluded that further studies with preexisting scores or new scores are needed to identify higher-risk patients.

Blood lactate level is a useful adjunctive marker for characterizing shock, and has been used to predict outcome and severity in numerous conditions [15]. Shah et al. performed a retrospective cohort study evaluating the usefulness of the initial lactate level at ED presentation in acute gastrointestinal hemorrhage [16]. A lactate level of >4 mmol/L was associated with a 6.4-fold increased odds of in-hospital mortality. El-Kersh et al. demonstrated the predictive value of lactate level on admission for in-hospital death in UGIB patients admitted to the ICU [17]. Lactate level adds to the predictive value of the pre-endoscopy Rockall score. Lee et al. evaluated several lactate parameters predicting the outcomes of non-variceal UGIB patients, and sought to develop a new scoring system by combining several lactate parameters and the AIMS 65 score [18]. On the basis of the above results, we found lactate level as a predictor in the new prognostic model. A cutoff value of lactate level of >2 mmol/L was chosen in the Sepsis-3 definition for septic shock [19]. Sammour et al. reported that trauma patients with lactate level >2 mmol/L showed higher mortality than those whose lactate level was <2 mmol/L [20].

Hemodynamic instability, such as shock, hypotension, and tachycardia, is one of the most predictive factors of UGIB severity for use in patient triage [2,8,21]. Rockall et al. reported that hypotension (SBP < 100 mmHg) and tachycardia (pulse ≥ 100) were calculated as 2 points and 1 point, respectively [7]. SBPs of 100–109, 90–99, and <90 mmHg were calculated as 1, 2, and 3 points, respectively, in the GBS. In major trauma patients, an SBP cutoff of <90 mmHg to identify high-risk patients has been challenged in several studies owing to compensatory mechanisms masking the process. Two triage scores indeed used higher cutoffs of 120 and 100 mmHg [22,23]. Haster et al. reported that SBP <110 mmHg was associated with increased mortality in patients with penetrating major trauma [24].

Fresh blood in nasogastric aspirate has been reported as a predictive factor for severity and outcome in many studies [7,25,26,27]. Corley et al. demonstrated that red blood in nasogastric lavage was an independent predictor of an adverse outcome (death, need for any operation, or recurrent hematemesis) [25]. Srygley et al. reported that a nasogastric lavage with red blood (summary likelihood ratio (LR), 3.1; 95% CI, 1.2–14.0) increases the likelihood of severe UGIB requiring urgent intervention [9]. Furthermore, a nasogastric lavage with blood or coffee grounds (LR, 9.6; 95% CI, 4.0–23.0) increases the likelihood that a patient has UGIB. However, a negative nasogastric lavage was seen in 72%, which provided little information and had a low sensitivity and poor negative likelihood ratio, limiting its usefulness in ruling out an upper gastrointestinal source of bleeding in patients with melena or hematochezia without hematemesis [28,29]. Furthermore, several studies reported that nasogastric aspiration failed to help clinicians in predicting the need for endoscopic hemostasis, did not improved in visualization of stomach at endoscopy [30,31,32]. Therefore, nasogastric tube insertion should be limited for patients who do not show obvious signs of dangerous clinical and laboratory findings and are well tolerated or cooperated.

The strength of our study is its large sample size and the inclusion of a homogenous cohort by analyzing only non-variceal UGIB patients with uniform treatment for UGIB at a single center. Our data were prospectively collected and the definitions of variables were predetermined. Although some variables with missing value existed, there were no unmeasured variables in terms of vital signs and laboratory results. In the final model, each component was objective (not using a subjective definition); hence, it will be easy to use in clinical practice. The components of our model are lactate level, blood in nasogastric tube, and SBP. These can be easily obtained because at ED presentation, lactate level can be assessed within 1 min with a blood gas analyzer. Our model is simple and the score can be calculated more rapidly than previous risk scores.

Several limitations should be addressed when interpreting our present results. First, it is a single-center study that reflects the characteristics of the hospital. External validation might be needed before our results can be generalized. Second, the episodes in the development and validation sets were divided in time order rather than being randomly selected. This could lead to differences in the characteristics of cohorts and treatments according to study period. However, most variables did not show a significant difference in vital signs, comorbidities, laboratory results, and outcomes. Third, our new risk score was developed using data of patients with normotension; hence, its utility for the overall non-variceal UGIB population is not clear. Fourth, we used hypotension as a surrogate measure of the clinically relevant outcomes of re-bleeding and mortality, while requirements for clinical intervention such as endoscopic hemostasis, transfusion, and radiologic intervention were outcomes in previous studies. Fifth, nasogastric tube irrigation should be used in limited patients due to low usefulness in predicting need for endoscopic hemostasis and uncomfortable procedure. Lastly, although patients with liver cirrhosis and advanced neoplasm were excluded, we did not exclude all possible factors affecting the lactate level, such as metformin use, advanced heart failure, acute kidney injury, and chronic kidney disease.

## 5. Conclusions

We have developed and validated a simple and objective risk score model that stratifies normotensive patients with non-variceal UGIB at a risk for the occurrence of hypotension. With our score model, non-variceal UGIB episodes were divided into different two classes and predicted the occurrence of hypotension. The accuracy testing of the new score showed that it has higher sensitivity and NPV than the GBS. This tool may allow for the triage of high- or low-risk patients at the point of care. However, further validation on an independent patient population would be warranted

## Figures and Tables

**Figure 1 jcm-08-00037-f001:**
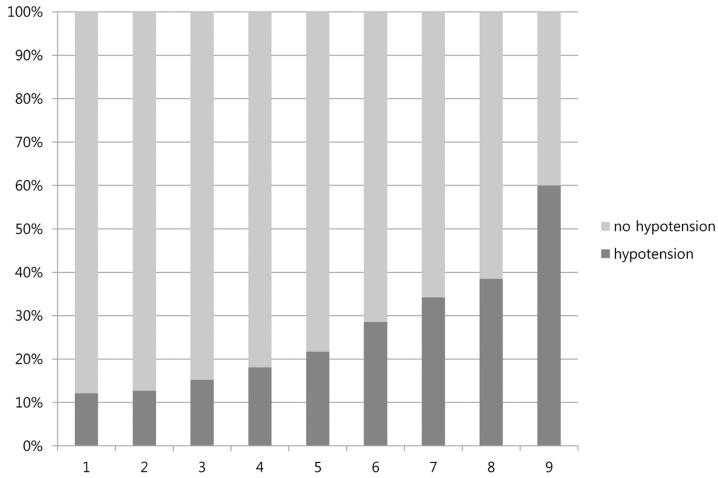
Observed occurrence of hypotension for the new scores in the development set.

**Figure 2 jcm-08-00037-f002:**
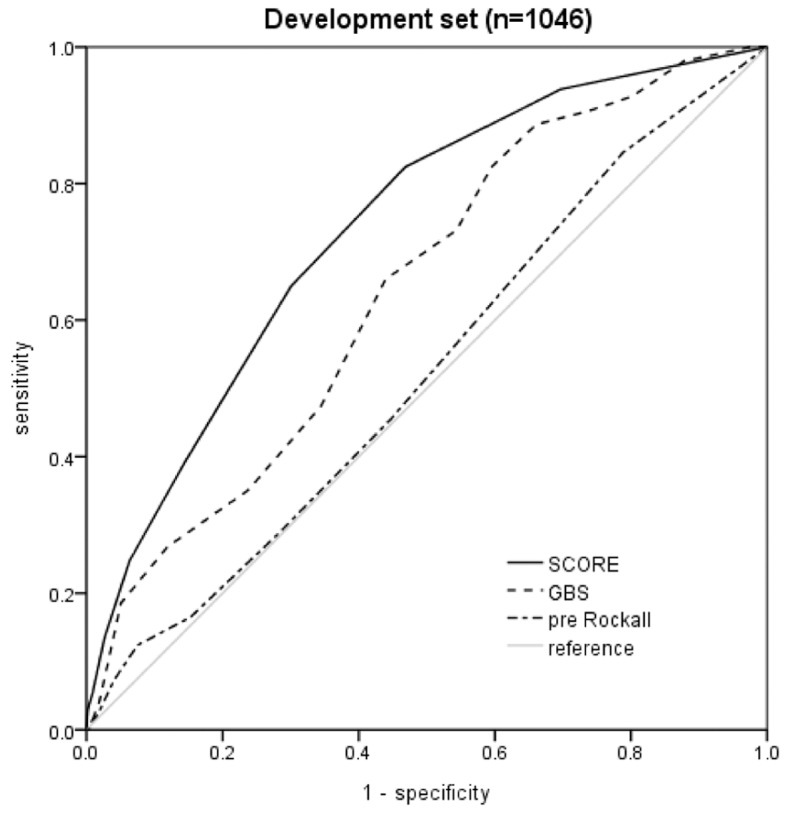
Comparison of the predictive ability of the new score with that of the GBS and the pre-endoscopy Rockall score in predicting the occurrence of hypotension in the development set. GBS, Glasgow–Blatchford score.

**Figure 3 jcm-08-00037-f003:**
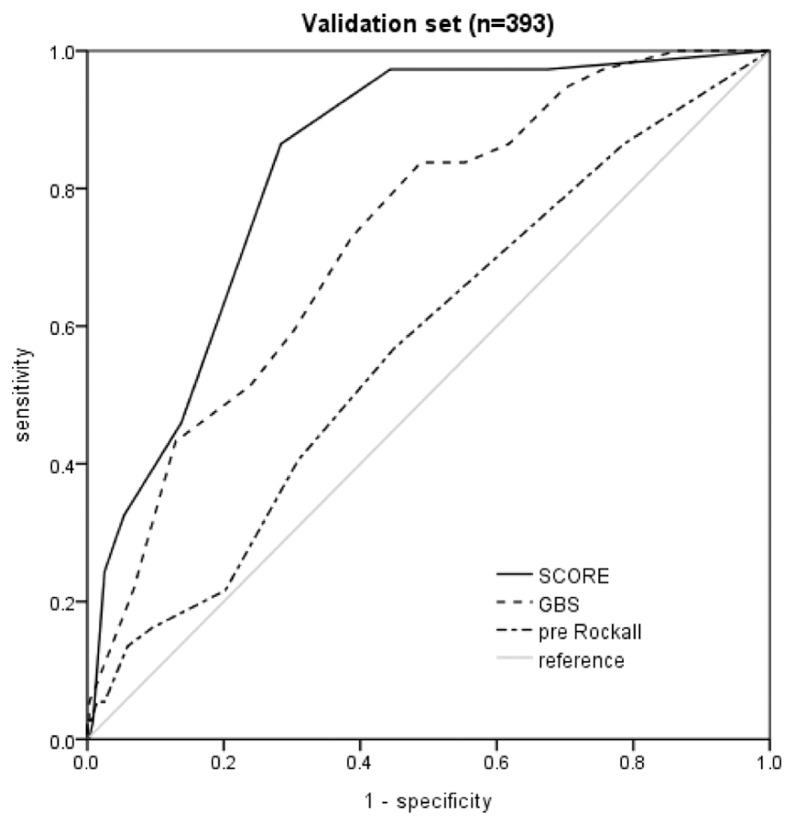
Comparison of the predictive ability of the new score with that of the GBS and the pre-endoscopy Rockall score in predicting the occurrence of hypotension in the validation set. GBS, Glasgow–Blatchford score.

**Table 1 jcm-08-00037-t001:** Univariable analysis model for predicting the occurrence of hypotension in the development set (2012–2015).

Characteristics	No hypotension Occurrence (*n* = 949)	Hypotension Occurrence (*n* = 97)	OR	95% CI	*p*
Demographics					
Age					
Mean ± SD	61.0 ± 16.7	60.9 ± 13.4	0.999	0.989–1.012	0.936
Male, *n* (%)	286 (30.1)	17 (17.5)	0.493	0.287–0.847	0.010
Comorbidities, *n* (%)					
Diabetes mellitus	197 (20.7)	26 (26.8)	1.398	0.869–2.250	0.168
Hypertension	358 (37.7)	38 (39.1)	1.063	0.693–1.632	0.779
Chronic liver disease	37 (3.9)	1 (1.0)	0.257	0.035–1.892	0.182
Coagulopathy	157 (16.5)	20 (20.6)	1.310	0.778–2.206	0.309
Ischemic heart disease	123 (12.9)	9 (9.3)	0.687	0.337–1.399	0.301
Heart failure	27 (2.8)	6 (6.2)	2.252	0.906–5.596	0.081
Neoplasm	124 (13.0)	18 (18.5)	1.516	1.516–2.616	0.135
CKD	81 (8.5)	7 (7.2)	0.833	0.374–1.859	0.656
Previous GIB history	144 (15.1)	17 (17.5)	1.188	0.684–2.064	0.541
COPD	21 (2.2)	2 (2.1)	0.927	0.214–4.016	0.920
Stroke	87 (9.2)	5 (5.2)	0.538	0.213–1.360	0.190
Associated symptom and signs					
Syncope	34 (3.6)	4 (4.1)	1.157	0.402–3.333	0.786
Melena on rectal examination	406 (42.7)	54 (55.6)	1.944	1.241–3.045	0.004
Fresh blood on nasogastric tube	90 (9.5)	23 (23.7)	3.082	1.827–5.201	<0.001
Mental change					
Yes	20 (2.1)	4 (4.1)	1.998	0.669–5.969	0.215
Drug history					
Antiplatelet agent	195 (20.5)	12 (12.3)	0.546	0.292–1.019	0.057
NSAIDs	34 (3.6)	3 (3.1)	0.859	0.259–2.850	0.804
Anticoagulation	71 (7.5)	10 (10.3)	1.421	0.707–2.856	0.323
Vital signs (mean ± SD)					
SBP (mmHg)	127.4 ± 20.2	113.9 ± 19.1	0.957	0.943–0.971	<0.001
DBP (mmHg)	77.5 ± 15.1	72.3 ± 17.4	0.976	0.961–0.991	0.001
Heart rate (/min)	91.2 ± 20.5	95.3 ± 19.5	1.010	0.999–1.020	0.063
Respiratory rate (/min)	19.7 ± 1.5	20.1 ± 2.2	1.133	1.017–1.262	0.024
Body temperature (°C)	36.5 ± 0.5	36.4 ± 0.6	0.647	0.427–0.979	0.039
Laboratory findings, median (mean, SD)					
Hemoglobin (g/dL)	10.4 ± 2.9	9.5 ± 2.7	0.900	0.837–0.967	0.004
Platelet count	230 ± 97	224 ± 92	0.999	0.997–1.002	0.587
(×10^3^/mm^3^)					
PT/INR (%)	90.1 ± 24.2	83.5 ± 24.5	0.990	0.982–0.998	0.012
PT/INR (s)	13.7 ± 10.9	14.3 ± 9.1	1.004	0.988–1.021	0.597
BUN (mg/dL)	30.5 ± 23.5	36.3 ± 23.9	1.008	1.001–1.016	0.023
Creatinine (mg/dL)	1.3 ± 1.6	1.3 ± 1.6	1.027	0.906–1.164	0.677
Albumin (g/dL)	3.3 ± 0.6	3.0 ± 0.6	0.505	0.365–0.698	<0.001
Lactate (mmol/L)	1.7 ± 1.3	2.6 ± 2.9	1.260	1.145–1.386	<0.001
Base deficit (mmol/L)	1.3 ± 3.9	0.1 ± 5.6	0.937	0.896–0.981	0.005
Risk scores, median (mean, SD)					
GBS	9.4 ± 3.6	11.2 ± 3.1	1.175	1.099–1.256	<0.001
Pre-endoscopy Rockall score	1.8 ± 1.7	2.0 ± 1.8	1.060	0.941–1.195	0.335

BUN, blood urea nitrogen; CI, confidence interval; CKD, chronic kidney disease; COPD, chronic obstructive pulmonary disease; DBP, diastolic blood pressure; GIB, gastrointestinal bleeding; GBS, Glasgow–Blatchford score, INR, international normalized ratio; NSAIDs, non-steroidal anti-inflammatory drugs: OR odds ratio; PT, prothrombin time; SBP, systolic blood pressure; SD, standard deviation.

**Table 2 jcm-08-00037-t002:** New prognostic models for predicting the occurrence of hypotension in non-variceal upper gastrointestinal bleeding (LBS).

Factors	Values	Points
Lactate (mmol/L)	2–3.9	1
	≥4	2
Blood in NG	Yes	2
SBP (mmHg)	<100	5
	100–109	4
	110–119	3
	120–129	2
Sum		9

NG, nasogastric tube; LBS, Lactate, Blood in nasogastric tube, Systolic blood pressure.

**Table 3 jcm-08-00037-t003:** Diagnostic performance in predicting the occurrence of hypotension in the development set (2012–2015; event *n* = 97/1046).

Cutoff Point	Number of Hypotension	Sensitivity	Specificity	PPV	NPV	AUC
≥1	91	93.8	30.3	12.1	97.9	0.620
≥2	88	90.7	36.5	12.7	97.5	0.638
≥3	80	82.5	53.1	15.2	96.7	0.682
≥4	63	64.9	69.9	18.1	95.1	0.680
≥5	38	39.1	85.5	21.7	93.2	0.627
≥6	24	24.7	93.6	28.5	92.4	0.586
≥7	13	13.4	97.3	34.2	91.6	0.541
≥8	5	5.1	99.1	38.4	91.0	0.515
≥9	3	3.1	99.7	60.0	90.9	0.513

AUC, area under the curve; NPV, negative predictive value; PPV positive predictive value.

**Table 4 jcm-08-00037-t004:** Test of accuracy of cutoff value in the validation set.

Score	Number of Hypotension	Sensitivity (%)	Specificity (%)	PPV (%)	NPV (%)	AUC
New score (≤2)	36	97.3	55.6	18.6	99.5	0.755
New score (≥7)	9	24.3	97.5	50.0	92.5	0.606
GBS (≥7)	35	94.5	30.1	12.3	98.1	0.612

AUC, area under the curve; GBS, Glasgow–Blatchford score; NPV, negative predictive value; PPV, positive predictive value.

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
