# Peer review of "Early Risk Score for Predicting Hypotension in Normotensive Patients with Non-Variceal Upper Gastrointestinal Bleeding"

_jcm, 2019, doi:10.3390/jcm8010037_

Reviewer 1 Report

Byuk Sung Ko et al developed a new score based on 3 parameters to predict the occurrence of hypotension in patients with nonvariceal upper gastrointestinal bleeding (NGIB). They found an acceptable accuracy with an AUC of 0.83. Main comments:

1) Need of transfusion and number of transfused blood units were not considered in this model. Please discuss, since they could be very interesting parameters in this setting.

2) Page 3 lines 118-119: please report INR and platelets ranges to define coagulopathy.

3) In the Methods section, please add the recruitment period of the study (one year? From when to…)

4) In tables 3 and 4 Authors should report the AUC values for each cutoff value.

5) One of the parameters used in this score is the presence of blood in nasogastric tube. However, ESGE guidelines (see Gralnek IM et al, Endoscopy 2015) do not recommend the use of routine nasogastric placement or lavage, with a strong evidence.

6) Glasgow Blatchford score (GBS) is validated to establish the timing of endoscopy and the need of urgent endoscopy and/or hospital admission. Therefore, GBS cannot be used to predict hypotension in comparison to this novel score, as reported in the present paper. Indeed, only one article has investigated GBS as a tool to predict hypotension (see Kim JS et al, Korean J Gastroenterol 2016).

Author Response

Byuk Sung Ko et al developed a new score based on 3 parameters to predict the occurrence of hypotension in patients with nonvariceal upper gastrointestinal bleeding (NGIB). They found an acceptable accuracy with an AUC of 0.83. Main comments:

1)     Need of transfusion and number of transfused blood units were not considered in this model. Please discuss, since they could be very interesting parameters in this setting

Response) Thank you for your comment. In the validation set, the AUC of new score for predicting need of RBC transfusion was 0.698 (95% CI, 0.647–0.750) while those of GBS was 0.838 (95% CI, 0.798.–0.878). The diagnostic performance of the pre-endoscopy Rockall score was 0.607 (95% CI, 0.552.–0.663). The median amount of RBC transfusion in low risk group (≤2 in new score) was 0.0 (0.0-2.0) while those in high risk group (≥3) was 2.0 (0.0-4.0) (P <0.001). The median amount of RBC transfusion of high risk (≥7) and low risk group classified by GBS were 0.0 (0.0-0.0) and 2.0(0.0-3.8), respectively. Significant difference was also observed between high risk and low risk group classified GBS (P <0.001). The following table is the comparison of accuracy of cutoff value for predicting need of transfusion in the validation set. Please check.

Supplementary Table 3. Test of accuracy of cutoff value for predicting need of red blood cell transfusion in the validation set

Score

Number of hypotension

Sensitivity (%)

Specificity (%)

PPV (%)

NPV (%)

AUC

New score (≤2)

76

62.7

66.7

67.0

62.4

0.647

GBS   (≥7)

181

88.7

62.4

71.8

83.7

0.756

GBS Glasgow-Blatchford Score, NPV negative predictive value, PPV positive predictive value

We added this issue at results section as follows:

“The AUC of new score for predicting need of red blood cell (RBC) transfusion was 0.698 (95% CI, 0.647–0.750) while those of GBS was 0.838 (95% CI, 0.798.–0.878). The diagnostic performance of the pre-endoscopy Rockall score was 0.607 (95% CI, 0.552.–0.663). The median amount of RBC transfusion in low risk group (≤2 in new score) was 0.0 (0.0-2.0) while those in high risk group (≥3) was 2.0 (0.0-4.0) (P<0.001). The median amount of RBC transfusion of high risk (≥7) and low risk group classified by GBS were 0.0 (0.0-0.0) and 2.0(0.0-3.8), respectively. Significant difference was also observed between high risk and low risk group classified GBS (P <0.001). We examined the accuracy using a cutoff value of 2 in the new score model. Sensitivity, specificity, PPV, and NPV of new score for predicting need of red blood cell were 62.7%, 66.7%, 67.0%, and 62.4%, respectively (Table S3)”

2)     Page 3 lines 118-119: please report INR and platelets ranges to define coagulopathy.

Response) In our hospital, normal range of prothrombin time as percent and second are 70%-140%, 10-13 second. Normal INR is between 0.8-1.3. Normal range of platelet counts is 150,000~350,000 (mm3). We added this issue at method section as follows.

“normal range of prothrombin time (%) and platelets are 70%-140% and 150,000~350,000/mm3”

3)     In the Methods section, please add the recruitment period of the study (one year? From when to…)

Response) We selected the development set from January 1, 2012, to December 31, 2015 for 4 years and the validation set from January 1, 2016, to April 30, 2017 for 1 year and 4 months. Please check at Development and validation of the new prognostic model part at methods section.

4)     In tables 3 and 4 Authors should report the AUC values for each cutoff value.

Response) We added the AUC values in table 3 and 4. Please check.

5)     One of the parameters used in this score is the presence of blood in nasogastric tube. However, ESGE guidelines (see Gralnek IM et al, Endoscopy 2015) do not recommend the use of routine nasogastric placement or lavage, with a strong evidence.

Response) Thank you for your valuable comment. We agree with opinion. We do not support routine use of nasogastric tube in UGIB. We think that nasogastric tube insertion should be limited for patients who do not show obvious signs of dangerous clinical and laboratory findings and are well tolerated or cooperated. We added this issue at discussion and limitation section as follows.

“Furthermore, several studies reported that nasogastric aspiration failed to help clinicians in predicting the need for endoscopic hemostasis, did not improved in visualization of stomach at endoscopy. Therefore, nasogastric tube insertion should be limited for patients who do not show obvious signs of dangerous clinical and laboratory findings and are well tolerated or cooperated”

“Fifth, nasogastric tube irrigation should be used in limited patients due to low usefulness in predicting need for endoscopic hemostasis and uncomfortable procedure”

6) Glasgow Blatchford score (GBS) is validated to establish the timing of endoscopy and the need of urgent endoscopy and/or hospital admission. Therefore, GBS cannot be used to predict hypotension in comparison to this novel score, as reported in the present paper. Indeed, only one article has investigated GBS as a tool to predict hypotension (see Kim JS et al, Korean J Gastroenterol 2016).

Response) We agree with your comment. Among pre-existing risk scores, GBS has been widely validated and used despite there are contradictory results with usefulness of GBS. We developed new risk score and comparison with pre-existing score was needed though GBS was not developed to predict the occurrence of hypotension. GBS has been validated to predict the low risk patients who can be discharged safely, but it have limitations in identifying high risk patients who will need embolization, surgical treatment or in identifying patients at a high risk for hemodynamic instability.

Reviewer 2 Report

Dear authors,

thank you for submitting your article “Early risk score for predicting hypotension in normotensive patients with non-variceal upper gastrointestinal bleeding” to Journal of Clinical Medicine. 

The authors conducted an observational cohort study in 1439 patients with non-variceal gastrointestinal bleeding and developed and validated a novel scoring system for risk assessment for UGIB. This novel score seems to have a higher sensitivity than the established risk scores (Glasgow-Blatchford Score and Rockall Score) for predicting the outcome of high-risk patients with UGIB. 

In my view this study is an important contribution in the research field and I would consider it for publication. I have the following questions to the authors:

1.     Intravenous PPI was applied pre-endoscopically. Which dose was used in the study?

2.     How were the patients sedated during endoscopy and did this affect the primary outcome (hypotension < 24h)?

3.     It seems that in the study population all patients received a nasogastric tube. Did patients with a variceal bleeding also receive a nasogastric tube?  

4.     Did the authors also test the score for patients with variceal bleeding?

Author Response

Dear authors,

thank you for submitting your article “Early risk score for predicting hypotension in normotensive patients with non-variceal upper gastrointestinal bleeding” to Journal of Clinical Medicine. 

The authors conducted an observational cohort study in 1439 patients with non-variceal gastrointestinal bleeding and developed and validated a novel scoring system for risk assessment for UGIB. This novel score seems to have a higher sensitivity than the established risk scores (Glasgow-Blatchford Score and Rockall Score) for predicting the outcome of high-risk patients with UGIB. 

In my view this study is an important contribution in the research field and I would consider it for publication. I have the following questions to the authors:

1.       Intravenous PPI was applied pre-endoscopically. Which dose was used in the study?

Response) The loading dose of pantoprazole (40mg 2vial) was administered intravenously and 2vial (mixed with 5% dextrose 100ml) of that was administered continuously at a rate of 10 ml/hour until there was no evidence of peptic ulcer disease.

2.       How were the patients sedated during endoscopy and did this affect the primary outcome (hypotension < 24h)?

Response) Thank you for your comment. We agree with your opinion. Unfortunately, we do not have the data related to sedative status. Hence it is difficult to differentiate effect of sedation on the occurrence of hypotension during endoscopy. However, we do not routinely sedate patients with bleeding for the risk of hemodynamic deterioration unless patients are too uncooperative. Hence, it does not seem that association of hypotension with sedative drug is apparent.

3.       It seems that in the study population all patients received a nasogastric tube. Did patients with a variceal bleeding also receive a nasogastric tube?  

Response) We excluded patients with liver cirrhosis as comorbidities in our study. However, small proportion (2.9%) of patients had variceal bleeding on endoscopy (Table S2). They did not know that they had liver cirrhosis until this hospital visit.

4.       Did the authors also test the score for patients with variceal bleeding?

Response) We did not test new score for patients with variceal bleeding.

Round  2

Reviewer 1 Report

In table S2 Authors reported 31 cases of variceal bleeding? Why? This study is indeed focused on non-variceal upper gastrointestinal bleeding. Please explain.

Author Response

In table S2 Authors reported 31 cases of variceal bleeding? Why? This study is indeed focused on non-variceal upper gastrointestinal bleeding. Please explain.

Response) Thank you for your review. Cirrhosis patients with variceal bleeding were excluded. However, there are a few patients whose endoscopic findings were diagnosed as variceal bleeding. Before enrollment, we identified patient’s comorbidities based on history or prior medical record rather than objective examinations such as endoscopy or CT scan. There are cases that patients did not know history of liver cirrhosis and variceal bleeding though they had liver cirrhosis actually. However, the number of cases was not frequent (2.9%), which did not seem to affect the result of our study.